# Lath Martensite Microstructure Modeling: A High-Resolution Crystal Plasticity Simulation Study

**DOI:** 10.3390/ma14030691

**Published:** 2021-02-02

**Authors:** Francisco-José Gallardo-Basile, Yannick Naunheim, Franz Roters, Martin Diehl

**Affiliations:** 1Max-Planck-Institut für Eisenforschung, Max-Planck-Straße 1, 40237 Düsseldorf, Germany; f.gallardo@mpie.de (F.-J.G.-B.); f.roters@mpie.de (F.R.); 2Department of Materials Science and Engineering, Massachusetts Institute of Technology (MIT), 77 Massachusetts Avenue, Cambridge, MA 02139, USA; naunheim@mit.edu; 3Department of Materials Engineering, KU Leuven, Kasteelpark Arenberg 44, 3001 Leuven, Belgium; 4Department of Computer Science, KU Leuven, Celestijnenlaan 200A, 3001 Leuven, Belgium

**Keywords:** packet, block, subblock, lath, steel

## Abstract

Lath martensite is a complex hierarchical compound structure that forms during rapid cooling of carbon steels from the austenitic phase. At the smallest, i.e., ‘single crystal’ scale, individual, elongated domains, form the elemental microstructural building blocks: the name-giving laths. Several laths of nearly identical crystallographic orientation are grouped together to blocks, in which–depending on the exact material characteristics–clearly distinguishable subblocks might be observed. Several blocks with the same habit plane together form a packet of which typically three to four together finally make up the former parent austenitic grain. Here, a fully parametrized approach is presented which converts an austenitic polycrystal representation into martensitic microstructures incorporating all these details. Two-dimensional (2D) and three-dimensional (3D) Representative Volume Elements (RVEs) are generated based on prior austenite microstructure reconstructed from a 2D experimental martensitic microstructure. The RVEs are used for high-resolution crystal plasticity simulations with a fast spectral method-based solver and a phenomenological constitutive description. The comparison of the results obtained from the 2D experimental microstructure and the 2D RVEs reveals a high quantitative agreement. The stress and strain distributions and their characteristics change significantly if 3D microstructures are used. Further simulations are conducted to systematically investigate the influence of microstructural parameters, such as lath aspect ratio, lath volume, subblock thickness, orientation scatter, and prior austenitic grain shape on the global and local mechanical behavior. These microstructural features happen to change the local mechanical behavior, whereas the average stress–strain response is not significantly altered. Correlations between the microstructure and the plastic behavior are established.

## 1. Introduction

Martensitic transformations are diffusionless phase transformations, i.e., they proceed by the cooperative and simultaneous movement of many atoms over distances less than an atomic diameter [1]. The most prominent example of a martensitic phase transformation is the formation of martensite from the austenitic (γ) phase in steels upon rapid quenching below the composition-dependent martensite start temperature M_s_. Carbon, present in solid solution in austenite, remains in solid solution in the new martensitic phase [1], which usually distorts the crystal lattice. The kinetics of the transformation and the morphology of the martensite are driven by the minimization of the strain energy in the presence of constraints from the neighboring microstructure which gives rise to elastic and plastic deformation [2].

In ferrous alloys, face-centered (fcc) austenite transforms–depending on the concentration of alloying elements and heat treatment–into three kinds of martensites with different crystal structures: $\alpha^{\prime }$ martensite (body-centered cubic, bcc, or body-centered tetragonal, bct), ϵ martensite (hexagonal close packed, hcp), and face-centered tetragonal (fct) martensite [3]. The $\alpha^{\prime }$ type martensite is most common in ferrous martensite and exists in the two major types plate martensite and lath martensite [4]. Lath martensite is formed in low-alloy steels, maraging steels, Interstitial Free (IF) steels, Dual Phase (DP) steels, and most other heat-treatable commercial steels. Hence, lath martensite is a material with overwhelming industrial significance [3,5,6].

The nature of martensite formation results in a distinct crystallographic orientation relationship between parent austenite and product martensite. Based on experimental observations on various steel grades, the following expressions have been proposed: {1 1 1}_γ_||{0 1 1}_$\alpha^{\prime }$_ and <1 0 1>_γ_||<1 1 1>_$\alpha^{\prime }$_ (Kurdjumov–Sachs, KS [7]), {1 1 1}_γ_||{0 1 1}_$\alpha^{\prime }$_ and <1 1 2>_γ_||<0 1 1>_$\alpha^{\prime }$_ (Nishiyama-Wassermann, NS [8,9]), and {1 1 1}_γ_||{1 0 1}_$\alpha^{\prime }$_ and <5 12 17>_γ_||<7 17 17>_$\alpha^{\prime }$_ (Greninger–Troiano, GT [10]). These models are still widely used, even though theoretical considerations predict deviations from orientation relationships in terms of low-indexed planes and directions [11]. Experiments by Morito et al. [12], Morito et al. [13] in Fe-C alloys with different carbon contents, in two Mn-containing steels, and in a maraging steel confirmed this: in all these alloys they found an orientation relationship that can be described as a near KS orientation relationship that deviates towards NW orientation relationship and is very close to GT orientation relationship. In view of the small angular differences between the different orientation relationships and the fact that all of them are just an approximation to the theoretically expected values, the features of lath martensite will be discussed in the following under the assumption of a KS orientation relationship. The 24 variants (V1–V24) defined by the KS model are given in Appendix A.

Lath martensite has a unique morphology that is shown in Figure 1 and Figure 2. The name-giving *lath* is named after its characteristic shape: Murata [14] reported a ratio of 30:7:1 for length, width, and thickness, respectively. This is in qualitative agreement with earlier findings by Wayman and Bhadeshia [15], who found laths with a length of 200 μm, a width of 4 μm, and a thickness of 0.3
μm. Multiple laths with similar crystallographic orientation form a block, a structure that is visible even with light optical microscopy [3]. Depending on the alloy composition, a block can be divided further into clearly distinguishable subblocks. The misorientation between two laths of the same subblock is reported to fall below 5${}^{\circ }$ [5]. Multiple blocks with the same habit plane form a packet [3,12,16]. The habit plane is the interface between austenite and martensite as measured on a macroscopic scale. For unconstrained transformations this interface plane is flat, but strain energy minimisation introduces faceted boundaries when the transformation is constrained by its surroundings. The interface between austenite and martensite as estimated on a macroscopic scale is defined as the habit plane. This interface is flat for unconstrained transformations but presents faceted boundaries when the transformation is obliged by its surroundings due to strain energy minimisation. Nevertheless, on a macroscopic scale, the habit plane is indistinguishable for the two cases. Habit planes are usually given as low indexed Miller indices even though the indices are irrational in lath martensite [1].

Packet and block size strongly depend on carbon content [16]. Morito et al. [12] have reported smaller packets and blocks for an increase in carbon content from 0.0026 wt% to 0.61 wt%. For low carbon steels however, packets consist of well developed parallel blocks [12]. A block is made up by two subblocks, each of which contains laths of a specific KS variant group [12]. Two subblocks belonging to the same block are found to have small misorientation angles of about 10${}^{\circ }$ between them, so only three blocks (i.e., pairs of subblocks) are possible in a packet. These pairs are V1–V4, V2–V5 and V3–V6 in the first packet [12] where the plane (1 1 1)${}_{\gamma }$ is aligned with (0 1 1)${}_{\alpha^{\prime }}$. The respective pairs in the remaining three packets, corresponding to the other three austenite planes, are given in Appendix A. Morito et al. [5] observed that the subblock boundaries were parallel to block boundaries in ultra-low carbon lath steel. They also found that the habit plane and the longest direction of the subblocks are close to (1 1 1)${}_{\gamma }$ and [1 0 1]${}_{\gamma }$, respectively. For high-carbon alloys containing around 0.8C wt%, packets and blocks are not distinguishable in optical micrographs [3]. Morito et al. [12] measured the size of the blocks in a high carbon alloy (0.61C wt%). These blocks had a width of a few micrometers and consisted of laths with a single variant. Six blocks with different orientations existed in a packet. Generally, the size of blocks and packets is also dependent on the grain size of the prior austenite and the quenching rate [18,19]. Morsdorf et al. [17] found also a slight decrease of lath size with increasing austenite grain size, although lath size is often considered as insensitive to austenite size [18,20,21]. Increasing the cooling rate and alloying to trigger partial formation of bainite are other ways to decrease the size of packets and blocks [22].

The mechanical properties of martensite strongly depend on morphology and crystallography of laths, (sub)blocks, and packets [17]. However, the complex microstructure of martensitic steels makes it difficult to identify the relevance of these individual features. In addition, neither a change in alloy composition nor in heat treatment allows to change one of these features independently of the others which is a prerequisite for systematic parameter studies. Since microstructure–property relationships are a prerequisite for mechanism-based alloy design, various experimental investigations have been conducted in order to establish such relationships: Li et al. [23] claimed that block width is the effective grain size for fatigue crack propagation in a lath martensite steel. Du et al. [24] quantified the strengthening due to block and subblock boundaries which act as a barrier for dislocation motion. A specific challenge of experiments on martensitic microstructures are the conflicting requirements posed by their multi-scale nature: high spatial resolution is needed to look at individual laths and a large field of view is needed to gain statistically valid insights. While scanning electron microscopy (SEM) basically achieves a good compromise between both requirements, the ability of understanding the micromechanical behavior from surface observations is strongly limited by the missing subsurface information [25,26]. Current 3D characterization techniques are, in contrast, not capable to resolve the smaller structures and cope with the lattice distortions introduced during quenching.

Understanding and predicting the plastic behavior of martensite is not only difficult due to its complex hierarchical structure, but also due to different deformation modes that have been reported: In an experimental study, Du et al. [27] observed apparent grain boundary sliding which might be responsible for unexpected high ductility of martensite. Maresca et al. [28] showed in a simulation study that retained austenite at interlath boundaries can significantly contribute to the deformation along the habit plane. Furthermore, it was reported that besides the usual <1 1 1> {1 1 0} and <1 1 1> {1 1 2} slip systems, the <1 1 1> {1 2 3} systems can be activated in martensite [29].

To systematically and independently investigate the effect of individual parameters, crystal plasticity simulations based on microstructures that include the experimentally observed hierarchy are a promising route. While microstructure evolution simulations, e.g., utilizing the phase field method, are the natural way to create these microstructures, their computational effort usually limits their application typically to single austenite grains [30]. Synthetic generation approaches that include selected features without relying on a physics-based simluation, are therefore a frequently used option: Briffod et al. [31] presented a computational study for the modeling of lath martensitic steels in two dimensions (2D), considering morphological and crystallographic features. For bainite, which has a similar crystallography as martensite, an approach to create three-dimensional (3D) synthetic microstructures by filling parent austenitic microstructure with packets has been presented by Osipov [32] and Osipov et al. [33]. Ghassemi-Armaki et al. [34] constructed a 3D Representative Volume Element (RVE) for modelling martensitic microstructures, which resolved the martensitic microstructure down to the level of individual blocks. Schäfer et al. [35] built a 3D RVE consisting of blocks to study the influence of strain ratio on fatigue crack initiation in lath martensite [35,36]. Maresca et al. [28] used experimental results of Morito et al. [12] to model plasticity of retained austenite in lath martensite at the scale of individual laths.

None of the simulation studies, however, considered the complete hierarchy of martensitic microstructures on the scale of multiple former austenitic grains. Here, an approach is presented that allows to generate martensitic microstructures with qualitatively realistic features that span the length scale from individual lath to multiple prior austenitic grains. Austenitic grains are transformed into martensite by dividing them into packets, blocks, subblocks, and laths subsequently. The microstructures generated with this approach are used to set up crystal plasticity simulations with the Düsseldorf Advanced Material Simulation Kit (DAMASK [37], https://damask.mpie.de/). A standard phenomenological formulation considering slip on <1 1 1> {1 1 0} slip systems is selected as the constitutive description. To enable high-resolution simulations, a fast spectral solver [38,39] is used.

The study is structured as follows: First, details of the martensitic microstructures generation approach are given in Section 2, including how the martensitic substructure is incorporated into the grains. The following section deals with the modelling framework used, including the numerical solution strategy, the crystal plasticity constitutive model, and the constitutive parameters. The simulation set up and the deformation conditions are described in Section 4. After that, results are presented and discussed in Section 5. The study finishes with a summary and an outlook on how to obtain more precise predictions of the mechanical behavior.

## 2. Generating Lath Martensitic Microstructures

The approach used in this study to generate martensitic microstructures allows to independently vary several of the characteristic features of lath martensite as outlined above in a systematic way and, hence, can be used to derive holistic microstructure–property relationships. The 3D prior austenitic microstructure is required as input. The procedure to generate a lath martensite microstructure is sketched in Figure 3. It is based on the assumption that the crystallographic orientation of the martensite is related to the parent austenite orientation by the KS relationship. The individual steps that are performed for each austenitic grain are the following:Packet generation: The austenitic grain (Figure 3a) is subdivided by two flat boundaries into three packets with approximately the same volume. Since no rules are established on how the packets geometrically partition the prior austenite grain, the boundaries are modelled to be perpendicular to each other. The resulting T-shaped grain boundary network is randomly rotated in space (Figure 3b).Subblock generation: For each packet, a different habit plane is selected that is parallel to a {111} plane of the austenitic grain. The packets are then subdivided into subblocks of thickness, $t_{\mathrm{subblock}}$, parallel to the habit plane (Figure 3c). According to Morito et al. [12], subblocks in low-carbon steels appear in pairs of crystallographic orientations. For example, the 6 variants of the ${\left(111\right)}_{\gamma }$ habit plane occur in the following pairs: V1–V4, V2–V5, and V3–V6. This variant selection is considered when assigning the crystallographic orientations. The order of the variants within a pair and the arrangement of the pairs is random, where for the former a direct repetition is disallowed.Lath generation: A Voronoi tessellation is performed in each subblock where each seed corresponds to one individual lath. The volume of the lath, $V_{\mathrm{lath}}$, is inversely proportional to the number of seeds. By distorting the resulting structure of equiaxed grains, laths with an average shape of length ($l_{\mathrm{lath}}$) > width ($w_{\mathrm{lath}}$) > thickness ($t_{\mathrm{lath}}$) are achieved. The longest direction, $l_{\mathrm{lath}}$, of the laths is aligned parallel to one of the <110> directions in the respective {111} plane, the shortest direction, $t_{\mathrm{lath}}$, is aligned normal to the plane. Each lath gets a crystallographic orientation assigned that deviates slightly from the nominal orientation according to the KS model (Figure 3d). More precisely, a random misorientation axis is chosen and the misorientation angle scatters randomly by a value between 0 and $\theta_{\max}$.

The martensitic microstructure is generated in a regular grid made of voxels and can be controlled by the following parameters:The thickness of the subblocks in the direction normal to the habit plane, $t_{\mathrm{subblock}}$. It is measured in units of length (UL) which corresponds to the side length of a voxel.The average volume of the lath, $V_{\mathrm{lath}}$, controlled via the number density of seeds used in the Voronoi tessellation. It is measured in units of volume (UV) which corresponds to the volume of a voxel, i.e., UL^3^.The average aspect ratio of the lath’s dimensions, $l_{\mathrm{lath}}\geq w_{\mathrm{lath}}\geq t_{\mathrm{lath}}$, controlled via the respective stretch factor.The maximum misorientation angle of the individual lath with respect to the nominal KS orientation, $\theta_{\max}$. It is measured in degrees (${}^{\circ }$).

In addition, random numbers are used for the following features due to the lack of their experimentally determined magnitudes:The rotation of the packet geometry.Sequence of variants within a subblock.Sequence of pairs within a block.Misorientaton distribution of the laths within the same subblock.

## 3. Modeling Framework

### 3.1. Numerical Solution Strategy

To spatially resolve individual laths in a martensitic grain aggregate and consider the volume of several parent austenite grains at the same time, a high resolution is required that enables to assign several computation points to each individual lath. In the present study, an advanced spectral solver [38,39] implemented in DAMASK [37] is used. It is an improved version of the method presented by Moulinec and Suquet [40] and Lahellec [41] for the solution of periodic mechanical boundary value problems. Since the stress equilibrium is calculated in Fourier space, the use of fast Fourier transforms (FFT) allows for a very memory- and time-efficient solution scheme.

### 3.2. Constitutive Model

At each material point x, the deformation gradient $F^{}\left(x\right)$ is multiplicatively decomposed into elastic and plastic parts as $F^{}=F_{e}^{}F_{p}^{}$. An anisotropic elastic stiffness C relates the elastic deformation gradient $F_{e}^{}$ to the second Piola–Kirchhoff stress by $S=C\;({F_{e}^{}}^{T}F_{e}^{}-I)/2$. The plastic velocity gradient $L_{p}={\dot{F}}_{p}^{}{F_{p}^{}}^{-1}$ is implicitly driven by S by virtue of the chosen plasticity model, which in the present study is an adoption of the phenomenological description of Hutchinson [42] for body-centered cubic crystals (for details see [43]). The microstructure is parametrized in terms of a slip resistance $g^{\alpha }$ on the twelve <1 1 1> {1 1 0} slip systems, indexed by α=1,…,12. These resistances evolve asymptotically from $g_{0}$ towards $g_{\infty }$ with shear $\gamma^{\beta }$ (β=1,…,12) according to the relationship
(1)$$\begin{array}{cc} {\dot{g}}^{\alpha } & ={\dot{g}}^{\alpha }\;h_{0}{\left|1-g^{\beta }/g_{\infty }\right|}^{a}\mathrm{sgn}\left(1-g^{\beta }/g_{\infty }\right)h_{\alpha \beta } \end{array}$$
with parameters $h_{0}$ and *a*. The hardening matrix $h_{\alpha \beta }$ describes the interaction between the different slip systems. Its values are 1.4 for non-coplanar and 1.0 for coplanar interactions. Given a set of current slip resistances, shear on each system occurs at a rate
(2)$$\begin{array}{cc} {\dot{\gamma }}^{\alpha } & ={\dot{\gamma }}_{0}{\left|\frac{\tau^{\alpha }}{g^{\alpha }}\right|}^{n}\mathrm{sgn}\left(\tau^{\alpha }\right) \end{array}$$
with ${\dot{\gamma }}_{0}$ as reference shear rate, $\tau^{\alpha }=S\;\cdot \;\left(m\alpha ⊗n^{\alpha }\right)$, and *n* the stress exponent. The plastic velocity gradient is then the sum of shear on all slip systems:(3)$$\begin{array}{c} L_{p}=\sum_{\alpha =1}^{N}{\dot{\gamma }}^{\alpha }\;m\alpha ⊗n^{\alpha }, \end{array}$$
where vectors mα and $n^{\alpha }$ are, respectively, unit vectors describing the slip direction and the normal to the slip plane of the slip system α and *N* is the number of (active) slip systems; ${\dot{\gamma }}^{\alpha }$ is the shear rate on that same system.

With this model the typically observed plastic deformation perpendicular to the habit plane can be achieved in the bulk as a combination of only two slip systems. However, the deformation is not confined to boundaries as it would be the case for boundary sliding, e.g., due to retained austenite films.

### 3.3. Constitutive Parameters

The parameters specifying the mechanical behavior of martensite are based on the parameter set determined by Tasan et al. [44] for a joint experimental-numerical analysis of stress and strain partitioning in DP steels. However, here the <1 1 1> {1 1 0} slip systems are exclusively used and the initial hardening rate ($h_{0}$), the initial resistance ($g_{0}$), and the saturation resistance ($g_{\infty }$) are adjusted to reproduce the stress–strain curve up to the ultimate yield stress of a fully lath martensitic microstructure obtained from a commercial steel (Dillidur 450) by AG der Dillinger Hüttenwerke. Its nominal chemical composition is given in Table 1. For the parameter identification, a microstructural map of size 400 μm × 400 μm was acquired with a step size of 0.35
μm by means of electron backscatter diffraction (EBSD). The raw data has been cleaned using the TSL software [45]. Points identified as retained austenite, i.e. approximately 2% of all indexed points, are assumed to be martentsite. This assumption–which significantly simplifies the parameter identification–is justified by the fact that the chosen material is designed as a fully martensitic steel. The resulting microstructure model, Figure 4a, is then loaded in uniaxial tension in rolling direction (RD) at the same rate as the corresponding experiment ($0.7\times 10^{-3}\;s^{-1}$) and up to the ultimate yield stress (at about 5% strain). By iterative adjustment of the material parameters, a good agreement between experimental and simulated curve could be achieved, see Figure 4b. The resulting constitutive parameters which are used throughout this study are given in Table 2.

## 4. Simulation Setup

In the following, the setup of the two simulation series used in this study is explained in detail. The aim of the first set of simulations is to investigate whether the presented approach for the generation of martensitic microstructures is capable of reproducing the behaviour of the experimental microstructure. Due to experimental limitations, only 2D simulations can be used for a direct comparison. Since it is known from previous studies that 3D setups are a prerequisite for the correct prediction of stress and strain partitioning at the grain scale [25,26,47], additional 3D simulations are performed to quantify the differences between 2D and 3D microstructure models. A second set of microstructures is created to investigate systematically the influence of martensite morphology and crystallography and prior austenitic grain shape on the microscopic and macroscopic behavior.

### 4.1. Simulations Based on Experimental Microstructures

To validate the presented approach, a comparison between results from simulations based on synthetic microstructures to results from simulations based directly on measured microstructures is performed. For this comparison, the following additional steps need to be taken: First, prior austenite grains are reconstructed from the 2D martensitic microstructure shown in Figure 5a. To this end, the ARGPE program in version 2.4 (March 2019) [48] is used. Quadruplets are used for the reconstruction method while all the other parameters remain as default. This resulted in a successful reconstruction of more than 95% of the pixels, see Figure 5b. Second, a 3D RVE is created using DREAM.3D [49] (Figure 5c). The statistical data from the 2D austenite reconstruction used for creating this synthetic 3D microstructure include grain size distribution, grain shape, and orientation distribution function (ODF). Since the available measurement does not provide any information about manufacturing-induced heterogeneity, no through-thickness variation is assumed. The texture index of the austenitic reconstruction was calculated to be 1.1 ([50]). The number of grains used for this reconstruction is a compromise between high spatial resolution and good statistical approximation of the weak/almost-random crystallographic texture: Using more grains, which necessarily translates into a lowered resolution, will result in a better match of the ODF. Using less grains, which would allow to increase the spatial resolution for the same computation time, would allow to finer resolve the features of the martensite. Finally, this microstructure is used to create a martensitic microstructure, see Figure 5d.

For the comparison, the following microstructures are used:Experimental microstructure: This is a direct 2D takeover of the measured crystallographic orientation of each of the 1143 × 1143 = 1,306,449 material points after cleaning out the retained austenite (Figure 4a). It is the same model that was used for the parameter adjustment (Section 3.3).3D RVEs: A regular grid of 256 × 256 × 256 = 16,777,216 material points with a 0.5
μm resolution that contains 86 equiaxed austenitic grains serves as the starting point. The values of the parameters used to create the martensitic structure from this microstructure are: $t_{\mathrm{subblock}}=15$ UL, $V_{\mathrm{lath}}=1400$ UV, $l_{\mathrm{lath}}:w_{\mathrm{lath}}:t_{\mathrm{lath}}=$ 9:3:1, $\theta_{max}=3{}^{\circ }$. A total of ten 3D RVEs are created using different random seeds.2D RVEs: These models are created by selecting a slice from a 3D model that contains 90 austenitic grains in a 600 × 600 × 50 = 18,000,000 grid. The 3D RVE used for slicing was created using the same parameters as for the 3D models. A total of three 2D RVEs with 600 × 600 = 360,000 points are used, choosing different slices from the same 3D model.

### 4.2. 3D Simulations with Systematically Varied Microstructural Features

To study the influence of lath microstructure on the global and local behavior, 3D RVEs with different values for the parameters of the martensite generation approach are created. For the study of one given parameter, the values of all other parameters remain the same as given in Section 4.1.

Lath volume: The value is set to $V_{\mathrm{lath}}=$ 320, 1400, and 4600 UV. Since subblocks are entirely filled with laths, a decrease in lath volume directly results in more lath per subblock and vice versa.Lath aspect ratio: Different lath shapes are created modifying $l_{\mathrm{lath}}$, $w_{\mathrm{lath}}$, and $t_{\mathrm{lath}}$. Rectangular cuboid-shaped laths are created with aspect ratio 9:3:1. Plate-shaped laths are created with aspect ratio 8:8:1. Rod-shaped laths are created with aspect ratio 5:1:1. Cube-shaped laths are created with aspect ratio 1:1:1.Scatter: The misorientation angle is chosen as $\theta_{\max}=$ 0, 3, and 5${}^{\circ }$. $\theta_{\max}=0{}^{\circ }$ represents a 3D RVE made only of subblocks since all laths in a subblock will have the same orientation. Limiting $\theta_{\max}<5{}^{\circ }$ is based on experimental evidence showing that the misorientation angle of a laths within a subblock does not exceed 5 ${}^{\circ }$.Subblock thickness: Subblock thickness is set to $t_{\mathrm{subblock}}=$ 8, 15, and 20 UL.

Additionally, a 3D RVE with elongated austenitic grains is created in order to study the effect of prior austenitic grain shape. Grains are elongated parallel to the rolling direction. The values of the parameters used to create this martensitic structure are the ones used in Section 4.1.

## 5. Results & Discussion

### 5.1. Simulations Based on Experimental Microstructures

#### 5.1.1. Average Stress–Strain Response

Stress–strain curves for the experimental microstructure, the 2D RVEs, and the 3D RVEs are given in Figure 6. For the latter two, a range is given to reveal the behavior of the different microstructures (2D: three different microstructures, 3D: ten different microstructures) that have been used. Values of yield stress ($\sigma_{\mathrm{yield}}$) and ultimate yield stress ($\sigma_{\mathrm{ultimate}}$) after reaching the final deformation are also given. Superscript and subscript indicates maximum and minimum value, respectively. The global mechanical response is almost insensitive to the microstructure used. 2D RVEs predict slightly higher stresses than the simulations based on the experimental microstructure, whereas 3D RVEs predict slightly lower stresses.

#### 5.1.2. Correlation of Stress and Strain Fields

For the statistical evaluation of the correlation between stress and strain, the state at the end of the loading, i.e., at about 5% strain, is considered. To quantify the correlation between selected mechanical fields, Pearson’s correlation coefficient *r* is computed. This coefficient measures the linear correlation between two data sets. Its value ranges from +1 to −1, where +1 indicates a total positive linear correlation, 0 no linear correlation, and −1 a total negative linear correlation. Table 3 contains the *r* values for selected correlations of strain in RD ($\varepsilon_{11}$), stress in RD ($\sigma_{11}$), von Mises equivalent strain ($\varepsilon_{\mathrm{vM}}$), and von Mises equivalent stress ($\sigma_{\mathrm{vM}}$). As expected for uniaxial loading, high values (0.97 to 0.98) for the correlation between $\varepsilon_{11}$ and $\varepsilon_{\mathrm{vM}}$ are found for all microstructures. In contrast, a low correlation (0.13 to 0.26) is found between $\sigma_{11}$ and $\sigma_{\mathrm{vM}}$. Moreover, a low negative correlation is found between $\varepsilon_{11}$ and $\sigma_{11}$ and a low positive correlation is found between $\varepsilon_{11}$ and $\sigma_{\mathrm{vM}}$.

When comparing the correlation coefficients between the experimental microstructure and the 2D RVEs, only negligible differences are seen for all four correlations. This indicates that the presented generation approach is capable to reproduce the deformation behaviour of complex hierarchical martensitic microstructures. For the comparison between 2D and 3D RVEs, significant differences are seen: $r(\sigma_{11}-\sigma_{\mathrm{vM}})$ is much higher for 3D microstructures then for 2D microstructures (0.26 vs. 0.14), while $r\left(\varepsilon_{11}-\sigma_{\mathrm{vM}}\right)$ is much lower. This finding is in agreement with earlier studies [25,26,51] which showed that the stress and strain partitioning between 2D and 3D microstructures is qualitatively different.

To further illustrate the differences of stress and strain distribution, heat-maps of $\varepsilon_{11}$ and $\sigma_{11}$ of each voxel are shown in Figure 7. Since all 2D and all 3D RVEs showed qualitatively identical results only one example is shown for either case. For the experimental microstructure (Figure 7a) and the 2D RVE (Figure 7b), a strong concentration at low strain levels can be seen. The strain distribution of the 3D RVE (Figure 7c) is, in contrast, almost symmetric around the average value.

#### 5.1.3. Micromechanics of 2D and 3D Models

Strain and stress distributions are plotted in Figure 8a,b, respectively. The two 3D RVEs with highest and lowest peak values are selected to show the range in the distributions whereas the results from all three 2D RVEs are given. The observed small differences between the RVEs of the same kind can be attributed to the randomness included in the creation process. It is confirmed that these differences are much smaller than the differences between microstructures of different kinds. To highlight the behavior at the right tail of the distribution, the relative number of points above a certain threshold is given as a bar chart in Figure 8c. This figure is based on the results of all simulations.

The experimental microstructure has a non-symmetric $\varepsilon_{11}$ distribution (see Figure 8a). The 2D RVEs reproduce this result, while for the 3D RVEs a more normal-like distribution can be seen. Regarding the stress distribution in Figure 8b, Gaussian distributions from experimental microstructure and 2D RVEs coincide while the 3D RVEs show a wider but also Gaussian distribution. Despite neglecting damage in the crystal plasticity simulations, the high strain and/or stress values shown in Figure 8c can be used to estimate the amount of damage. DAMASK has already been used successfully for determining a correspondence between the high-equivalent-strain regions and the onset points of strain localization [52]. Correct predictions of the right-sided tail of the stress and strain distributions are, therefore, of special importance. This holds especially for $\sigma_{\mathrm{vM}}$, which is usually used as indicator for damage initiation in brittle materials. The 2D RVEs show the same characteristics as the experimental 2D microstructure, suggesting once more that the created 2D microstructures correctly reproduce the behaviour of the experimental one. The 3D RVEs show significant differences in the number of high values of $\varepsilon_{11}$ and $\varepsilon_{\mathrm{vM}}$ in Figure 8. The different behavior in 3D reveals that correlation sites of of high stress/strain with damage initiation sites in 2D simulations can be misleading as the realistic situation in 3D is fundamentally different.

In the following, spatially resolved results are discussed (see Figure 9). For $\varepsilon_{\mathrm{vM}}$ as well as for $\varepsilon_{11}$ (not shown), an X-shaped distribution is seen. This X-pattern originates from the maximization of the resolved shear stress under ±45${}^{\circ }$ to the RD. Locally, highest $\varepsilon_{\mathrm{vM}}$ values reach the 8-fold of the prescribed mean deformation. Stress values are strongly dependent on the crystallographic texture. Like the experimental microstructure, 2D RVEs and 3D RVEs show the development of bands of intense plastic deformation generally oriented at about ±45${}^{\circ }$ from the tensile direction, in which the plastic strain reaches up to eight times the prescribed mean deformation.

Spatially resolved results from a 3D RVE are shown in a surface normal to RD (see Figure 10), where no X-pattern is obtained. This is helpful to analyze the preference of strain localization at certain boundaries. Strain localizes mainly around prior austenitic grain and packet boundaries. Voxels with $\epsilon_{\mathrm{vM}}>0.13$ are colored in red and can be directly located adjacent to these high-angle boundaries. Plastic strain concentrations occur less frequently at block and subblock boundaries as can be seen in Figure 10. At lath boundaries, concentration of plastic deformation is rarely found. In Figure 10 two arrows are plotted to distinguish two zones. In zone 1, a block boundary with low strain and high stress values is shown, indicating that high values of strain cannot necessarily be associated to high values of stress. In zone 2, a packet boundary where strain and stress localizes is shown. It is shown that stress localizes also at boundaries, but the preference is not as clear as for the strain. One reason for this is that the stress is strongly dependent on the crystallographic orientation.

### 5.2. 3D Simulations with Systematically Varied Microstructural Features

The macroscopic stress–strain curve (not shown in this study) is almost insensitive to variation of the parameters that are used in this study to control the microstructure. The comparison of heat maps similar to those shown in Figure 7 revealed also no obvious differences in the stress strain partitioning (not shown in this study). Hence, in order to enable an in-depth analysis, the fraction of points where stress and strain exceeds certain threshold values at the final deformation state are given in Table 4.

It can be seen that smaller magnitudes for $\varepsilon_{\mathrm{vM}}$ are found for the smallest lath volume. This is because the lath volume is inversely proportional to the number of laths within a subblock. By increasing the number of laths, a more random distribution and a better partitioning amongst repetitive orientations within a subblock is obtained. It also implicates a greater amount of low-angle misorientation lath boundaries. More of these boundaries results in less high values. No clear trend between high values of $\sigma_{\mathrm{vM}}$ and the lath volume was found. Regarding lath aspect ratio, high values of $\varepsilon_{\mathrm{vM}}$ follow the order: 8:8:1 (1.508) > 5:1:1 (1.498) > 9:3:1 (1.457) > 1:1:1 (1.456). It is not the same as for high values of $\sigma_{\mathrm{vM}}$: 5:1:1 (1.271) > 9:3:1 (1.246) > 8:8:1 (1.179) > 1:1:1 (1.12). Still, the lowest values for $\varepsilon_{\mathrm{vM}}$ and $\sigma_{\mathrm{vM}}$ are seen for an aspect ratio of 1:1:1, i.e., cube-shaped laths. Hence, this–very unrealistic–lath shape can be considered as the one which has the lowest propensity for damage initiation. Varying the scatter results in the following order of high values for $\varepsilon_{\mathrm{vM}}$ for and $\sigma_{\mathrm{vM}}$, respectively: 0${}^{\circ }$ (1.510) > 3${}^{\circ }$ (1.498) > 5${}^{\circ }$ (1.475) and 5${}^{\circ }$ (1.261) > 3${}^{\circ }$ (1.246) > 0${}^{\circ }$ (1.225). This reveals that a higher lath misorientation results in higher stress partitioning since lath boundaries contribute in addition to grain, packet, block, and subblock boundaries to the microstructural heterogeneity. For higher misorientation, in contrast, strain localization at lath boundaries becomes less probable. To study this behavior in further detail, half of a prior austenite grain is shown in Figure 11. Strain localization at both boundaries of the light green subblock is seen for $\theta_{\max}=0{}^{\circ }$. For $\theta_{\max}=5{}^{\circ }$, the strain distribution is less pronounced at the left boundary. In addition, a region of high plastic deformation parallel to the right one can be seen in the green subblock to the right. This region is not located in the vicinity of a (sub)block boundary but at a lath boundary inside a subblock. Since laths can have a misorientation up to $5{}^{\circ }$ with respect to the nominal KS orientation, high angle misorientations can be created which behave similar to other types of boundaries. Imposing lath boundaries and increasing their misorientation angles can, hence, preferentially shift the regions of high strains from subblock or block boundaries to lath boundaries.

A consistent behavior is found for $\varepsilon_{\mathrm{vM}}$ and $\sigma_{\mathrm{vM}}$ when increasing the subblock size: large subblocks result in higher partitioning of strain and stress (see Table 4). Differences can, however, be seen in the distribution of $\varepsilon_{11}$ and $\sigma_{11}$ values (Figure 12). While a slight and consistent shift to lower values of stress can be seen for smaller subblocks in Figure 12b, no clear trend is obvious for the strain distribution. For $t_{\mathrm{subblock}}=8$, more subblocks are created and therefore a better randomization of the ODF is achieved. This results in low numbers of high values of $\sigma_{\mathrm{vM}}$ (see Table 4). This choice introduces more block and subblock boundaries compared to $t_{\mathrm{subblock}}=20$. Again, strain localization is distributed to more points. This makes high values of $\varepsilon_{\mathrm{vM}}$ less probable (see Table 4). *Subblock size* is the only parameter that drastically changes the spatially resolved results. In Figure 13, results for $t_{\mathrm{subblock}}=20$ and $t_{\mathrm{subblock}}=8$ are shown. For $t_{\mathrm{subblock}}=20$, prior austenitic grain boundary 1 (yellow arrow) shows strain localization while for $t_{\mathrm{subblock}}=8$ there is none. Prior austenite grain boundary 2 (yellow arrow) shows the opposite situation. In the figure, other strain localization zones at prior austenite grain boundaries can be seen for $t_{\mathrm{subblock}}=20$ and not for $t_{\mathrm{subblock}}=8$ and vice versa. This means that the subblock size parameter does not only affect the local strain distribution inside a prior austenite grain, which was expected, but also at the prior austenitic grain boundaries. As for the stress distribution, 2 grains are marked in Figure 13. For $t_{\mathrm{subblock}}=8$, grain 1 shows many gradients, which enables differentiating between the different subblocks. For $t_{\mathrm{subblock}}=20$, grain 1 shows a very homogeneous distribution of the stress. Grain 2 shows very similar stress distributions for both cases, where the stress in the bottom packet is more pronounced than in the upper one.

Finally, the results using equiaxed and elongated austenitic grains are compared. The macroscopic stress–strain curve again showed no difference between both cases. Figure 14 summarizes the most important results: Elongated grains narrow the strain distribution (Figure 14a), while no influence of the austenite grain shape on the stress distribution was observed (Figure 14b). As previously shown, strain localizes primarily at prior austenitic grain and packet boundaries inclined by $\pm 45{}^{\circ }$ with respect to the RD, where the resolved shear stress is maximized. The 3D RVE with elongated grains in the tensile direction has more prior austenitic grain boundary surface parallel to RD compared to the 3D RVE with equiaxed grains. The probability of finding a prior austenitic grain boundary inclined with an angle of $\pm 45{}^{\circ }$ is smaller, which causes a narrower Gaussian distribution. Less potential high-strain points makes the Gaussian distribution more narrow and the drop to zero faster. This effect can also be seen in Figure 14c, where high values of $\varepsilon_{11}$ and $\varepsilon_{\mathrm{vM}}$ are less frequently observed for the elongated case. The distribution for stress in RD are very similar (Figure 14b). High values of $\sigma_{11}$ and $\sigma_{\mathrm{vM}}$ were reached for both the elongated and equiaxed case. The 3D RVE with elongated grains in the RD is therefore more resistant to damage initiation.

## 6. Summary and Outlook

In this study, a parametrized approach is presented to generate microstructures of lath martensite from 2D or 3D representations of austenite polycrystals. The mechanical behavior of the generated microstructures is then investigated by means of crystal plasticity simulations.

The comparison of the results obtained from the experimentally measured microstructure and the 2D RVEs show a good quantitative agreement. Stress and strain distributions and their characteristics change significantly when 3D microstructures were used. For all microstructures, strain localization is primarily found in the vicinity of high-angle boundaries, prior austenite grain and packet boundaries, followed by block and subblock boundaries and finally at lath boundaries. High values of strain are not directly correlated with high values of stress. Stress localizes also at boundaries, but no preference for specific boundary types was detected.

The presented approach allows to systematically vary features of the lath martensite microstructure to study their influence. This approach is used to investigate the influence of individual microstructural features on the global as well as local stress and strain distribution with high statistical confidence by simulation. In contrast, experimental investigations often lack statistical validity. Microstructural features happen to change only the local mechanical behavior but not the average stress–strain response. The frequency of high values of strain and/or stress is calculated and used as a damage initiation indicator. Based on this criterion the most suitable microstructure would consist in cube-shaped and small subblocks. Whether a higher scatter inside the subblock is beneficial or not depends on whether the damage initiation is controlled by $\varepsilon_{\mathrm{vM}}$ and/or $\sigma_{\mathrm{vM}}$. Elongated prior austenite grains are also beneficial in comparison to equiaxed grains. It is shown that the local preferences of strain localization can change with the scatter. Strain localization at lath boundaries is favored by a high orientation scatter of the laths within a subblock. While the volume and shape of the laths do not alter the spatially resolved results for stress and strain, scatter can slightly change them. The subblock size and the prior austenitic grain shape show the biggest influence on stress and strain partitioning.

While the current simulations allow to investigate trends for the relation between microstructure and properties, computational limitations do not allow for a quantitatively correct description of the lath martensite morphology. A higher spatial resolution for a given number of austenitic grains would allow to approach the extreme aspect ratios of martenstitic lath and, hence, increase the predictive quality of the presented approach. Another aspect of the current study that needs to be improved for better predictions is the employed constitutive model. Since slip on <1 1 1> { 1 0 0} systems is not the only deformation mechanism observed in lath martensite, the use of models that incorporate additional slip systems (<1 1 1> {1 1 2} and <1 1 1> {1 2 3}), grain boundary sliding [27], damage [53], or size effects [54] is expected to improve the predictions. Finally, the retained austenite typically found in martensitic steels should be considered because it enables grain boundary sliding and, hence, can have a large influence on the overall deformation kinetics [28]. This can be done by explicit incorporation of the austenitic phase into the microstructure model or by means of a suitable constitutive description [55].

## Figures and Tables

**Figure 1 materials-14-00691-f001:**
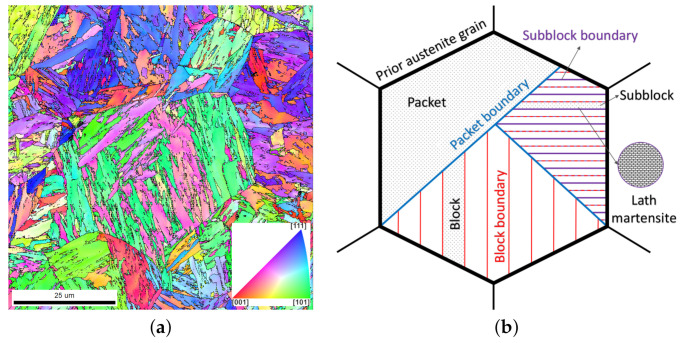
(**a**) Map of the Inverse Pole Figure (IPF) along the normal/out-of-plane direction of a Fe-0.13C-5.1Ni (wt%) model alloy. The heat treatment consisted of austenization at 900${}^{\circ }$ for 5 min and subsequently quenching in water to obtain a fully martensitic microstructure [17]. (**b**) Schematic of the hierarchical microstructure of lath martensite.

**Figure 2 materials-14-00691-f002:**
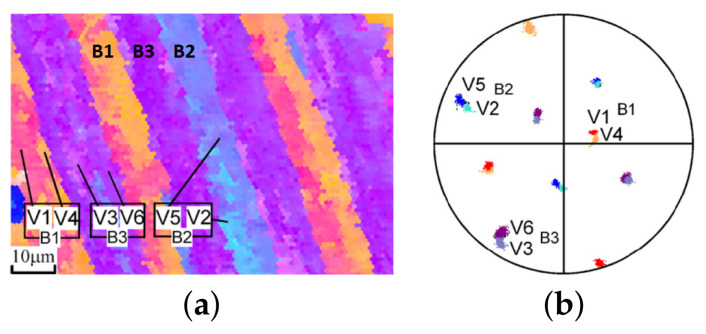
(**a**) Inverse Pole Figure (IPF) map of a 18Ni-8Co-5Mo maraging steel and (**b**) 100 pole figure, showing the crystal-orientation. Blocks are made of 2 subblocks with a low-angle misorientation of about 10 degrees (V1–V4, V2–V5 and V3–V6). Color legend is given in Figure 1a. Adapted with permission from ref. [13]. Copyright 2006 Elsevier.

**Figure 3 materials-14-00691-f003:**
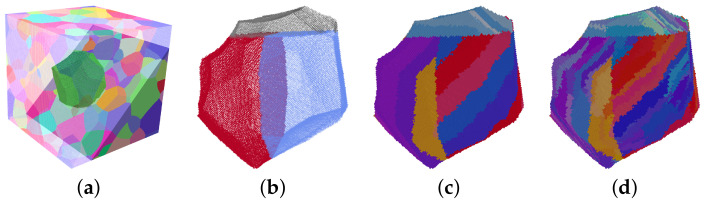
Creation of a martensitic microstructure from an austenitic microstructure: (**a**) Austenitic microstructure with highlighted grain. (**b**) Division of the grain in 3 packets. (**d**) Creation of the subblocks. (**e**) Creation of the laths by means of Voronoi tessellation. Legend for Inverse Pole Figure (IPF) in (**a**,**c**,**d**) is given in Figure 1a.

**Figure 4 materials-14-00691-f004:**
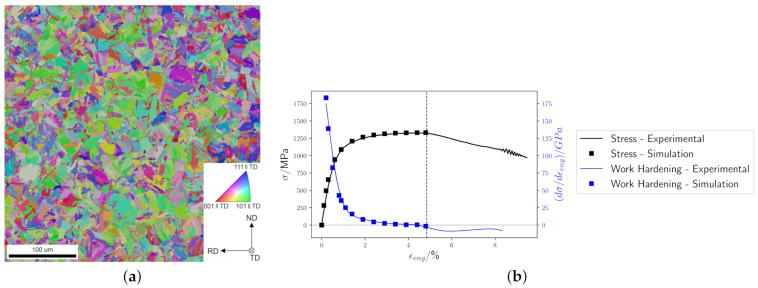
(**a**) Inverse Pole Figure (IPF) map of the experimental microstructure after cleaning. (**b**) Experimental and simulated stress–strain and hardening rate curves.

**Figure 5 materials-14-00691-f005:**
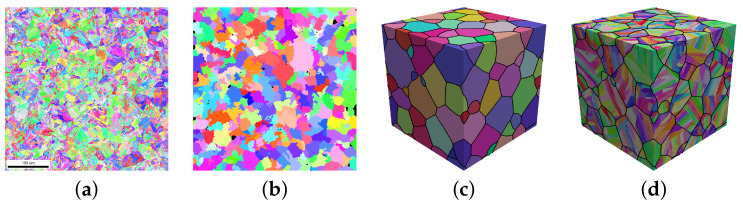
Inverse Pole Figure (IPF) maps, color legend is given in Figure 2a: (**a**) Raw experimental microstructure (includes retained austenite). (**b**) Prior austenitic microstructure. Black color means no indexing was possible. (**c**) Synthetic austenitic microstructure in 3D. (**d**) Martensitic microstructure. Black lines in (**c**,**d**) represents austenite grain boundaries.

**Figure 6 materials-14-00691-f006:**
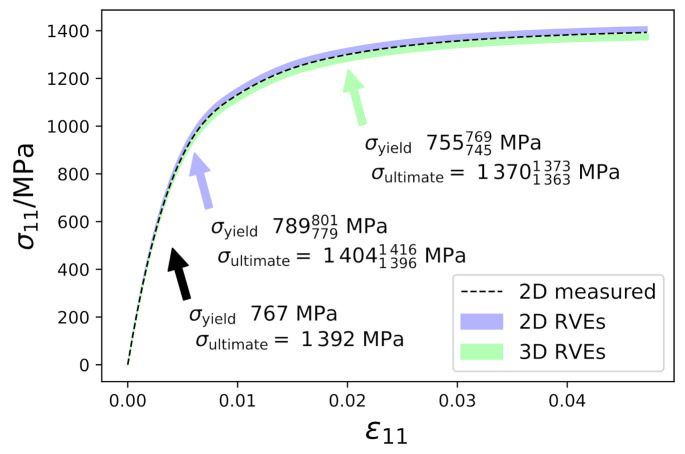
Stress–strain curves in Rolling Direction (RD).

**Figure 7 materials-14-00691-f007:**
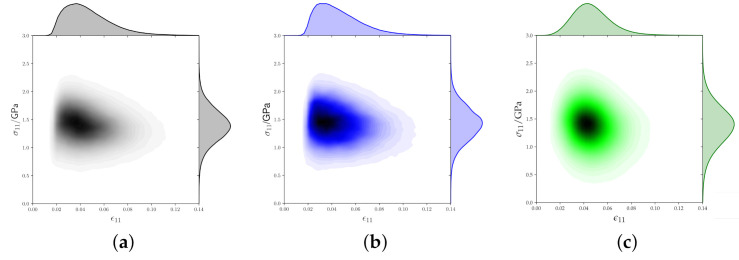
Heatmaps showing stress versus strain distributions in Rolling Direction (RD). (**a**) Measured microstructure. (**b**) 2D Representative Volume Element (RVE). (**c**) 3D RVE.

**Figure 8 materials-14-00691-f008:**
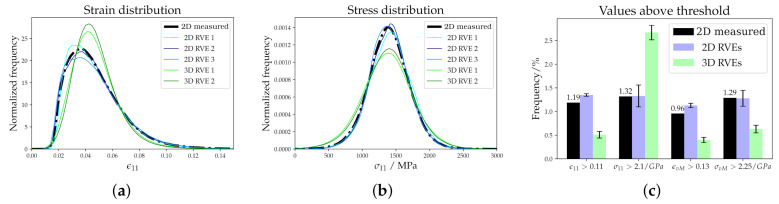
Statistical evaluation: (**a**) Strain and (**b**) stress distributions. (**c**) Number of points with values above a certain threshold. The threshold values are selected to obtain around 1% frequency for the data of the experimental microstructure.

**Figure 9 materials-14-00691-f009:**
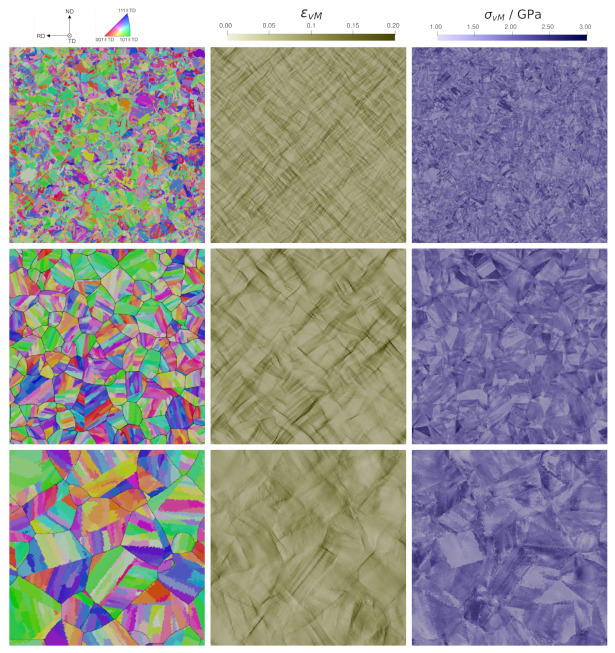
Spatially resolved results for the experimental microstructure of size 400 μm^2^ (**top**), a 2D Representative Volume Element (RVE) of size 300 μm^2^ (**middle**) and a 3D RVE of size 128 μm^3^ (**bottom**).

**Figure 10 materials-14-00691-f010:**

Spatially resolved results of a 3D RVE in a section normal to the rolling direction: Inverse Pole Figure (IPF), equivalent strain, and equivalent stress.

**Figure 11 materials-14-00691-f011:**
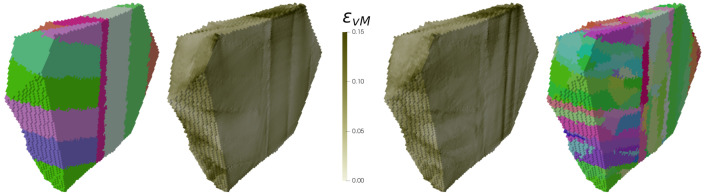
Inverse Pole Figure (IPF) (for color legend see Figure 2a) and equivalent strain of a part of the generated microstructure corresponding to 3D RVEs with $\theta_{\max}=0{}^{\circ }$ (**left**) and $\theta_{\max}=5{}^{\circ }$ (**right**). Two packets belonging to a prior austenite grain are distinguishable.

**Figure 12 materials-14-00691-f012:**
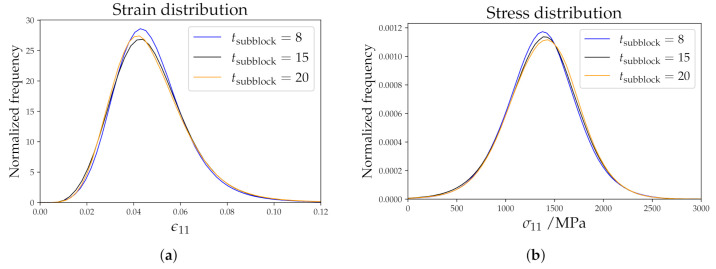
Distributions of strain (**a**) and stress (**b**) in the rolling direction for different subblock sizes.

**Figure 13 materials-14-00691-f013:**
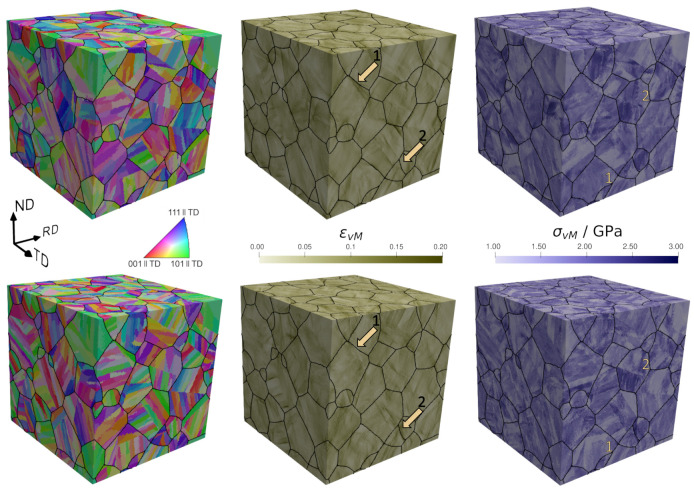
Inverse Pole Figure (IPF) and equivalent strain and stress distribution for an RVE with $t_{\mathrm{subblock}}=20$ (**top**) and $t_{\mathrm{subblock}}=8$ (**bottom**).

**Figure 14 materials-14-00691-f014:**
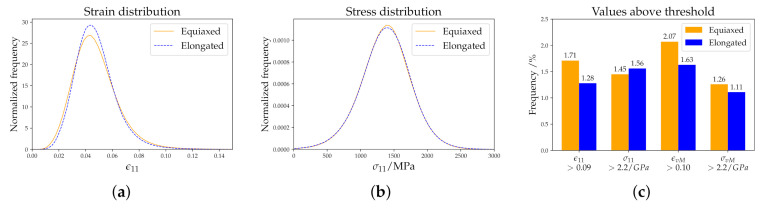
Strain (**a**) and stress (**b**) distributions. Number of points with values above a certain threshold (**c**). The threshold values are selected to obtain around 1–2% frequency.

**Table 1 materials-14-00691-t001:** Nominal chemical composition of the martensitic steel in wt%. Conventional quenching was used to produce lath martensite.

C	Si	Mn	P	S	Cu	Al	Nb	Mo	Ni	Cr
≤0.25	≤0.70	≤1.60	≤0.025	≤0.010	≤0.30	≤0.03	≤0.05	≤0.50	≤0.80	≤1.50

**Table 2 materials-14-00691-t002:** Adjusted material parameters, based on [44,46].

Property	Symbol	Value	Unit
Elastic constant	$C_{11}$	417.4	GPa
Elastic constant	$C_{12}$	242.4	GPa
Elastic constant	$C_{44}$	211.1	GPa
Initial resistance	$g_{0}$	160.0	MPa
Saturation resistance	$g_{\infty }$	555.0	MPa
Initial hardening rate	$h_{0}$	90.0	GPa
Reference shear rate	${\dot{\gamma }}_{0}$	10^−3^	s^−1^
Stress exponent	*n*	20	
Strain hardening exponent	*a*	2.0	

**Table 3 materials-14-00691-t003:** Pearson’s coefficient for stress and strain measures.

	$r\left(\varepsilon_{11}-\varepsilon_{\mathrm{vM}}\right)$	$r\left(\sigma_{11}-\sigma_{\mathrm{vM}}\right)$	$r\left(\varepsilon_{11}-\sigma_{11}\right)$	$r\left(\varepsilon_{11}-\sigma_{\mathrm{vM}}\right)$
2D measured	0.98	0.13	−0.15	0.12
2D RVEs	0.98	0.14	−0.16	0.13
3D RVEs	0.97	0.26	−0.13	0.04

**Table 4 materials-14-00691-t004:** Fraction of points above the threshold value for stress and strain. The threshold values are selected to obtain around 1–2%.

	Lath Aspect Ratio ($l_{\mathrm{lath}}$:$w_{\mathrm{lath}}$:$t_{\mathrm{lath}}$)				Lath Volume ($V_{\mathrm{lath}}$/UV)			Scatter ($\theta_{\max}$/${}^{\circ }$)			Subblock Size ($t_{\mathrm{subblock}}$/UL)		
	8:8:1	5:1:1	9:3:1	1:1:1	320	1400	4600	0	3	5	8	15	20
$f(\varepsilon_{\mathrm{vM}}>0.105)$/%	1.508	1.457	1.498	1.456	1.470	1.498	1.533	1.510	1.498	1.475	1.248	1.498	1.599
$f(\sigma_{\mathrm{vM}}>2.2\mathrm{GPa})$/%	1.179	1.271	1.246	1.121	1.213	1.246	1.225	1.225	1.246	1.261	1.234	1.246	1.481

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
