# Peer review of "Lath Martensite Microstructure Modeling: A High-Resolution Crystal Plasticity Simulation Study"

_materials, 2021, doi:10.3390/ma14030691_

Round 1

Reviewer 1 Report

The manuscript is written well, the corresponding research was conducted well, therefore I do not have any doubts  recommending publication. 

Author Response

We thank the reviewer for the very positive reception of our work.

Reviewer 2 Report

Title: Lath Martensite Microstructure Modeling: A High-Resolution Crystal Plasticity Simulation Study

Author(s): F.J. Gallardo-Basile, Y. Naunheim, F. Roters, M. Diehl

This manuscript presents a systematic study of microstructural features (lath volume, aspect ratio, scatter, subblock thickness) and representative volume element (RVE) reconstruction techniques for crystal plasticity simulations of a lath martensitic steel. The martensitic steel is characterized using simple tensile and EBSD experiments, while the DAMASK package is used for the crystal plasticity calculations. The study shows a good correlation between the simulation and experimental measurements. The results, discussion, and conclusions drawn are clear and to the point.

Overall, the manuscript is well-written and executed. Minor edits for grammatical errors and typos are provided. There are also some minor comments that the authors need to address.

Please see my detailed comments below:

Detailed Comments:

  1. Introduction
  1. Page 2 – Line 52: Should be “martensite” instead of “Martensite”.
  2. Page 2 – Line 52: “…has a unique morphology THAT is shown …” Please note the edit
  3. The introduction is well written. However, the state about martensite transformation from austenite is not entirely accurate: “It occurs for rapid quenching below the composition-dependent martensite start temperature M_s”. As the authors acknowledge in their conclusion, retained austenite can be present in the material. Depending on the chemistry, the austenite can transform from the TRIP effect, as was presented in the following references

Hu, X.H., Sun, X., Hector Jr, L.G. and Ren, Y., 2017. Individual phase constitutive properties of a TRIP-assisted QP980 steel from a combined synchrotron X-ray diffraction and crystal plasticity approach. Acta Materialia, 132, pp.230-244.

Park, T., Hector Jr, L.G., Hu, X., Abu-Farha, F., Fellinger, M.R., Kim, H., Esmaeilpour, R. and Pourboghrat, F., 2019. Crystal plasticity modeling of 3rd generation multi-phase AHSS with martensitic transformation. International Journal of Plasticity, 120, pp.1-46.

Connolly, Daniel S., Christopher P. Kohar, Waqas Muhammad, Louis G. Hector Jr, Raja K. Mishra, and Kaan Inal. "A coupled thermomechanical crystal plasticity model applied to Quenched and Partitioned steel." International Journal of Plasticity (2020): 102757.

Please include these references and any additional references along with a statement about deformation-induced austenite to martensite. 

  1. Modeling framework
  1. Page 6 – Line 178: It appears that the hardening matrix H_{\alpha\beta} fully assumes self-hardening and latent hardening are identical. This is an important assumption that should be discussed with a sentence or two.
  2. Page 6 – Line 182: Please report the chemistry of the material. It is important for the follow-up comments.
  3. Page 6 – Line 192: The material consists of approximately 2% austenite. Yet, it appears that only one set of material constants are used (which the reviewer assumes are for the lath martensite). Please explicitly state the assumptions about the treatment of the austenite phase in this current work. How is it modeled and what is the justification for such an approach.
  4. Page 6 – Line 192: With respect to the austenite phase, please comment if there were any observed transformation effects. 
  1. Simulation setup 
  1. Page 7 – Line 201: “…from previous studies that …” Note the edit removal of the “,”.
  2. Page 7 – Line 209: “For this comparison[,] the following …” Please note the edit.
  3. Page 7 - Line 213-215: “Second, a 3D RVE is created using DREAM.3D [50] (Fig. 5c). Relevant statistical data from the 2D austenite reconstruction used for creating this synthetic 3D microstructures including grain size distributions, grain shape, and orientation distribution function (ODF).” There are a lot of assumptions that are made here with respect to through-thickness heterogeneity (i.e., texture, phase, grain size, etc.). Please add some commentary about the assumptions that are made.
  4. Page 7 – Line 222: “Experimental microstructure”: Although it is clear through the explanation, the subsequent discussion and graphs can become quite confusing as they are not actual experimental measurements and can mislead a reader (especially if they are reading it quickly). In essence, the authors directly took the EBSD map and utilized it in their study. Perhaps the authors could change “Experimental” to “Direct EBSD”, or “Measured EBSD” or something more fitting, and edit Figures 6 – 8 and Table 2 accordingly.
  5. Page 9 – Line 287 – 288: “The expected small differences between the RVEs of the same kind can be attributed to the randomness included in the creation process, which can be …” Please note the edit(s).

Author Response

Please find our comments in the attached pdf.

Reviewer 3 Report

The authors present a study on crystal plasticity modelling of lath martensite microstructures. The topic is timely and of high scientific and industrial interest. While the creation of a 3D-martensitic microstructure from an austenitic microstructure is promising and a spectral solver is a suitable tool to model the deformation in such microstructures, the connection to the experimentally observed deformation mechanisms in lath martensite are missing. Therefore the model in its current form has no merit in predicting the plastic deformation in lath martensite. 

Here are my general comments and more detailed comments are given directly within the manuscript:

  • The introduction is excessively detailed and reminds more of a review paper - too many facts are introduced that are not at all relevant for the paper. Those could easily be handled by referring to other work.
  • The generation of a martensitic microstructure from a model austenitic microstructure is well done. Here the simplifications should be highlighted, such as the use of a simplified habit plane ({111} instead of {557}) and manual packet generation without considering strain minimization. While I think the current method is good enough for conducting the analysis, it could be further improved by the generation of a martensitic microstructure from for instance a phase field model to take into account the variant selection based on strain minimization across different austenite grains.
  • The major issue with this study is that the plasticity model does not reflect the experimentally observed deformation mechanisms in lath martensite. The only implemented mechanism of plasticity is slip in {110}<111>, whereas studies of the recent years clearly have indicated that lath martensite preferably deforms by boundary sliding and only partially accommodates deformation by slip on {110}<111>, see [C. Du, R. Petrov, M.G.D. Geers, J.P.M. Hoefnagels, Mater. Des. 172 (2019) 1–12.] as well as [C. Tian, D. Ponge, L. Christiansen, C. Kirchlechner, Acta Mater. 183 (2020) 274–284]. .

    As a consequence, the comparisons between modeling results of the experimental microstructure and the 2D and 3D modelled microstructures can be done relative to each other to evaluate the merit of using the modelled microstructures for simulation. However, none of the simulations can be expected to reproduce the deformation that is actually occurring in a lath martensite microstructure. Therefore the conclusions taken from the parameter study in section 5.2 are not transferable to real life, but only valid within the wrongly assumed dominating plasticity mechanism within the model. The only actual connection to experimental data is a good fit between the experimental and simulated tensile curves, which can be achieved by adapting the material parameters to almost any deformation system. Therefore there are several ways out:
    • You could highlight that your model is not valid to predict the actual plastic deformation occurring in lath martensite and limit the scope of the paper only to creation of model microstructures for crystal plasticity simulations of lath martensite without attempting to predict anything.
    • You could implement boundary sliding as an actual deformation mechanism into your model. To the best of my knowledge, this is unfortunately not straight forward.
    • You could conduct in-situ tensile EBSD measurements to directly compare the modelled local deformation with the measured local deformation to have an actual experimental validation. This would show whether your model actually has predictive capabilities. I however anticipate that, in agreement with literature, you will find boundary sliding to be a dominating effect, which will lead you to either implementing this mechanism in your plasticity code or stating that your model cannot predict deformation of lath martensite.

The fact that you suggest the implementation of further slip-systems as future work suggests to me that you have treated lath martensite as ferrite with a more complex and hierarchical morphology and texture. Literature has however shown that the mechanical response and the plastic deformation mechanisms are quite different for ferrite and lath martensite.

Author Response

Please find our reply in the attached pdf.

Round 2

Reviewer 3 Report

I am generally pleased with the response by the authors. However, the major criticism I had in my initial comments has not been addressed at all. I urge the authors to give a justification for their assumption of the plastic deformation mechanism implemented in their model, up to now it is merely an assumption without any justification. See my detailed comments to the authors' replies below:

We agree that the introduction is long and removed some details that are not relevant for the current work. However, since the two other reviewers positively commented on the handed-in version of the introduction, we limited the removal to aspects of martensite formation that are not relevant for the present study.

Don’t get me wrong, not all, but most of the content in the introduction is relevant for the work, but I think you would be better off by moving all the details on lath martensitic microstructures and orientation relationships into the next section. Only at line 104 you start introducing the motivation for your work and the chosen approach to advance the current state of the art. This is however just my recommendation to improve the structure of your paper, if you prefer the current form then that is ok with me as well.

The simplification is already mentioned in the introduction (end of 4th paragraph) and is now further improved. Note that in [1], the {1 1 1} plane is given as a good approximation for habit planes in low-alloy steels.

The simplification is absolutely reasonable and I am happy with the changes you implemented.

We agree that physics-based generation of RVEs is desirable. However, all approaches known to us result in much simpler microstructures then the presented approach or are limited to a single austenitic grain.

That makes sense, maybe you want to comment on this in the manuscript?

Missing constitutive ingredients have been mentioned already in the outlook of the assubmitted version of the manuscript. We have further detailed this (last paragraph of the manuscript) and also mentioned the characteristics of the model in the introduction and in the abstract.

I have seen these listed limitations, but none of them mentioned boundary sliding. I can also see that you have not mentioned this in the revised version. It seems unreasonable to me to spend a lot of effort into describing the exact morphology of the lath martensite microstructure based on extensive literature research (which is well done) and then simply assume a slip system in section 3.3 without any experimental or literature justification. I would like you to argue in section 3.3, based on experimental results on lath martensite plasticity from literature, which plasticity mechanisms should ideally be implemented in your model. You then have all freedom to argue that implementation of additional slip systems or a boundary sliding mechanisms are not feasible to implement and that you therefore exclusively consider the ⟨1 1 1⟩{1 1 0} system. Currently it just looks like you are avoiding the discussion rather than being upfront about your motivations and limitations.

Indeed, implementing grain boundary sliding is a task that is not doable in the scope of this study. However, we are working on the implementation of a model for martensite taking the effect of retained austenite films into account [2]. This model is a continuum approximation to sliding of martensite grains on austenite film.

There would be no shame in having this included in the manuscript and argue that you have therefore not considered it. In this way you could already pitch your future work as well.

Unfortunately, in-situ EBSD measurements are not suitable for comparison between experiments and simulations because the sub-surface microstructure is unknown. Therefore, the simulation needs to be 2D which makes any comparison useless [3]. Nevertheless, we did simulations based on the initial stage of an in-situ experiment presented in [4] and could reproduce the dominant strain localization when considering only plastic slip as the deformation mechanism, see Fig. 1. In a second simulation, where an EBSD map containing more former austenite grains is used, the agreement is less striking [5] – this is expected due to the missing sub-surface microstructure. In summary, we have no evidence from these simulations that the predictions for martensite plasticity are particularly bad in comparison to other materials when using a standard crystal plasticity model.

I understand the experimental limitations you describe and agree that adding experimental results to this manuscript would be out of scope.

PS.:
I think a verb is missing in this sentence of the abstract:
“Several laths of nearly identical crystallographic orientation group together to blocks […]”
